# Hepatitis E Virus Infection in Voluntary Blood Donors in the Russian Federation

**DOI:** 10.3390/v16040526

**Published:** 2024-03-28

**Authors:** Ilya A. Potemkin, Karen K. Kyuregyan, Anastasia A. Karlsen, Olga V. Isaeva, Vera S. Kichatova, Maria A. Lopatukhina, Fedor A. Asadi Mobarkhan, Anna G. Zlobina, Andrey V. Zheltobriukh, Ksenia A. Bocharova, Vera V. Belyakova, Svetlana V. Rassolova, Nadezhda V. Ivanova, Sergey A. Solonin, Alexey I. Bazhenov, Mikhail A. Godkov, Mikhail I. Mikhailov

**Affiliations:** 1Laboratory of Viral Hepatitis, Mechnikov Research Institute of Vaccines and Sera, 105064 Moscow, Russia; axi0ma@mail.ru (I.A.P.); a.carlsen@yandex.ru (A.A.K.); isaeva.06@mail.ru (O.V.I.); vera_kichatova@mail.ru (V.S.K.); marialopatukhina@yandex.ru (M.A.L.); 1amfa@bk.ru (F.A.A.M.); michmich2@yandex.ru (M.I.M.); 2Laboratory of Molecular Epidemiology of Viral Hepatitis, Central Research Institute of Epidemiology, 111123 Moscow, Russia; 3Belgorod Blood Center, 308007 Belgorod, Russia; ag.zlobina31@list.ru (A.G.Z.); nmspkb@mail.ru (A.V.Z.); 4Medical Faculty, Belgorod State National Research University, 308015 Belgorod, Russia; doctor.bocharova@mail.ru; 5Gavrilov Moscow Blood Center, Moscow Health Department, 125284 Moscow, Russia; karnas@mail.ru (V.V.B.); svetlanamishakina772@gmail.com (S.V.R.); nadya220iv@yandex.ru (N.V.I.); 6Sklifosovsky Research Institute for Emergency Medicine, Moscow Health Department, 129090 Moscow, Russia; soloninsa@sklif.mos.ru (S.A.S.); albazhenov@yandex.ru (A.I.B.); mgodkov@yandex.ru (M.A.G.)

**Keywords:** blood donor, hepatitis E virus, viremia, transfusion-transmitted infection

## Abstract

Transfusion-transmitted hepatitis E virus (HEV) infection is an increasing concern in many countries. We investigated the detection rate of HEV viremia in blood donors in Russia. A total of 20,405 regular repetitive voluntary non-renumerated blood donors from two regions (Moscow and Belgorod) were screened for HEV RNA using the cobas^®^ HEV test in mini-pools of six plasma samples. Samples from each reactive pool were tested individually. The average HEV RNA prevalence was 0.024% (95% CI: 0.01–0.05%), or 1 case per 4081 donations. No statistically significant differences in HEV RNA prevalence were observed between the two study regions. The PCR threshold cycle (Ct) values ranged from 25.0 to 40.5 in reactive pools, and from 20.9 to 41.4 in reactive plasma samples when tested individually. The HEV viremic donors had different antibody patterns. Two donor samples were reactive for both anti-HEV IgM and IgG antibodies, one sample was reactive for anti-HEV IgM and negative for anti-HEV IgG, and two samples were seronegative. At follow-up testing 6 months later, on average, four donors available for follow-up had become negative for HEV RNA and positive for anti-HEV IgG. The HEV ORF2 sequence belonging to HEV-3 sub-genotype 3a was obtained from one donor sample. The sequencing failed in the other four samples from viremic donors, presumably due to the low viral load. In conclusion, the HEV RNA detection rate in blood donors in Russia corresponds with data from other European countries, including those that implemented universal donor HEV screening. These data support the implementation of HEV RNA donor screening to reduce the risk of transfusion-transmitted HEV infection in Russia.

## 1. Introduction

Hepatitis E virus (HEV), or Paslahepevirus balayani, is a single-stranded RNA virus that is a member of the Hepeviridae family. HEV is currently classified into eight genotypes (HEV-1 to HEV-8) [1]. The viral genotype largely determines the epidemiology and predominant transmission route of HEV infection. Genotypes HEV-1 and HEV-2 are strictly anthroponotic and associated with outbreaks and sporadic infections, mainly water-borne, in countries with poor sanitation and limited access to high-quality drinking water [2]. Genotypes HEV-3 to HEV-8 infect various mammal species; HEV-3 and HEV-4 circulate mainly in wild and domestic pigs and deer and are the main cause of autochthonous zoonotic infection in humans in industrial countries [3]. HEV-3 and HEV-4 infections in animals and immunocompetent humans are often asymptomatic and self-limited [4]. HEV-5 and HEV-6 have been identified in wild boars and are not isolated from humans so far, while HEV-7 and HEV-8 has been identified in camelids, with HEV-7 confirmed to cause zoonotic infections in humans [5,6]. In addition to Paslahepevirus balayani, other members of the Hepeviridae family can cause infections in humans. For instance, rat hepatitis E virus, or Rocahepevirus ratti, is reported to be an emerging cause of acute infections in humans in recent years [7].

Along with water-borne and food-borne transmission, transfusion-transmitted HEV (TT-HEV) infection has also been described [8]. The latter may be of a great concern for immunosuppressed patients, such as recipients of hematopoietic stem cell or solid organ transplants who are at risk of chronic HEV infection, resulting in persistent liver inflammation and cirrhosis [9,10,11]. The risk of TT-HEV infection is associated with HEV viremia in asymptomatic donors; the viremic period has recently been estimated to be up to 88 days [12]. Based on donor prevalence data and the estimated risk of TT-HEV infection, a number of European countries and Japan have introduced universal donor screening for HEV RNA in recent years through mini-pools (MP) or individual donations (ID) [12,13]. 

In the Russian Federation, hepatitis E has been a notifiable disease since 2013. The annual incidence rates of hepatitis E in the country in that time have been low, ranging from 0.08 to 0.12 per 100,000 population. Seroprevalence studies have shown that HEV infection in the territory is much more widespread than previously assumed from the official registration data. The average prevalence of anti-HEV IgG antibodies in the Russian Federation is 4.6%, with significant variation between regions [14]. In the European part of the Federation, there are regions with increased intensity of HEV circulation, accompanied by increased incidence rates and a high prevalence of anti-HEV antibodies, such as the Belgorod region, a region in the European part of the Russian Federation located 600 km south of Moscow, close to the border with Ukraine [15]. The data on the prevalence of anti-HEV IgM antibodies indicating current or recent infection in the general population of Russia also suggest widespread prevalence of infection; the value varies from 0.2% to 2.8%, depending on the surveyed region [14]. TT-HEV infection appears to be relevant for the Russian Federation, as the prevalence of anti-HEV IgM antibodies in blood donors was previously shown to be as high as 2.8–4.5%, depending on the region [16]. However, screening of blood donors for HEV RNA is not implemented in Russia so far, and the data on HEV RNA prevalence in blood donors are absent. The primary aims of this study are to assess the prevalence of active HEV infection in voluntary blood donors based on the detection of viral RNA in plasma and to evaluate the significance of HEV RNA screening in blood donors in Russia.

## 2. Materials and Methods

### 2.1. Donor Samples

A total of 20,405 regular repetitive voluntary non-renumerated blood donors were recruited in 2022–2023 for the study, including 14,533 donors from Moscow (at Moscow Blood Center) and 5872 donors from the Belgorod region (at Belgorod Blood Center). All samples were obtained from different donors, i.e., a single sample was obtained from each donor. All donors were adults aged 18 years and older. No study-specific informed consent forms were obtained, as all recruited donors signed the standard informed consent before donation, which contains a statement on the possibility that donor samples may be used for additional population-based studies. The study design and procedure were approved by the local ethics committee of the Mechnikov Research Institute of Vaccines and Sera (approval no. 4, dated 18 April 2022).

EDTA plasma samples were collected once during the routine screening of donors, between October 2022 and September 2023. All EDTA plasma samples had a volume ≥ 3 mL, were centrifuged within 24 h after blood collection, and were stored at +4 °C for no more than 3 days or at <−18 °C for up to 30 days before HEV testing.

Donors who appeared to be positive for HEV RNA were invited for a follow-up visit an average of 6 months after the initial positive result.

### 2.2. HEV Testing

All donor plasma samples were tested for HEV RNA in mini-pools of 6 (MP-6) with the cobas^®^ HEV kit in the cobas 6800 System (Roche Diagnostics, Mannheim, Germany). According to kit insert, the 95% limit of detection (95% LoD) of the test is 18.6 HEV RNA international units (IU) per mL (95% CI: 15.9–22.6 IU/mL) when samples are tested individually, which corresponds to a 95% LoD of 2.05 log10 IU/mL for MP-6. Pooling was performed using the Hamilton Microlab STAR Liquid Handler system (Hamilton Company, Reno, NV, USA). Samples from each reactive pool were tested individually, as well as samples obtained at the follow-up visit. Each positive result was confirmed by repeated individual testing.

All individual reactive plasma samples were subjected to supplementary amplification of the open reading frame 2 (ORF2) fragment of the HEV genome, as described elsewhere [14]. Both initial and follow-up plasma samples from donors positive for HEV RNA were tested for anti-HEV IgM and IgG antibodies using commercial enzyme immunoassay (EIA) tests: DS-EIA-ANTI-HEV-M and DS-EIA-ANTI-HEV-G, respectively (Diagnostic Systems, Nizhny Novgorod, Russia).

All HEV RNA and anti-HEV antibody testing was performed according to the manufacturer’s instructions in the respective kits.

### 2.3. HEV Sequencing and Phylogenetic Analyses

An amplified HEV genome fragment 350 bp in length was purified from agarose gel using a QIAquick Gel Extraction Kit (QIAGEN, Hilden, Germany) and sequenced on a 3130 Genetic Analyzer automatic sequencer (ABI, Foster City, CA, USA) using a BigDye Terminator v3.1 Cycle Sequencing Kit, following the manufacturer’s protocol. Sequences were aligned using MEGA 11 software [17]. A phylogenetic tree was built based on a 300 nt fragment of the HEV ORF2 region (corresponding to nt positions 5996–6295, numbered as strain M73218) using PhyML 3.0 under a GTR model with SPR tree correction (http://www.atgc-montpellier.fr/download/papers/phyml_spr_2005.pdf, assessed on 27 March 2024) and an aLRT SH-like test. Tree annotation was performed using TreeAnnotator v.1.8.4 for 1000 replicates and visualized using FigTree v.1.4.3.

### 2.4. Statistical Analysis

Statistical analysis was performed using GraphPad and included the calculation of proportion (%) for prevalence data with a 95% confidence interval (95% CI) and the assessment of the significance of differences in mean values between groups using Chi-square with Yates correction, with a significance threshold of *p* < 0.05.

## 3. Results

The data on HEV RNA prevalence in blood donors are shown in Table 1. HEV RNA was detected in 3 out of 14,533 plasma samples from Moscow donors, which is a ratio of 1 case per 4844 donations, and in 2 out of 5872 plasma samples from Belgorod donors, or a ratio of 1 case per 2936 samples. No statistically significant differences in HEV RNA prevalence were observed between the two study regions. When the data from the two regions were combined, the HEV RNA prevalence in blood donors was 1 case per 4081 donations (0.024%, 95% CI: 0.01–0.05%).

HEV viremic donors did not report any health problems at the time of sampling and had normal ALT levels, which are measured as part of the routine donor checkup. The demographic characteristics of reactive donors and detailed PCR results are presented in Table 2. Two HEV viremic donors were women and three were men. PCR threshold cycle (Ct) values ranged from 25.0 to 40.5 in MP6, and from 20.9 to 41.4 in reactive plasma samples when tested individually. The average difference between Ct values in MP6 and ID testing was 3.4 (minimum: 2.3; maximum: −4.1).

HEV viremic donors had different anti-HEV antibody patterns at the initial testing, as shown in Table 3. Two donor samples were reactive for both IgM and IgG antibodies, one sample was reactive for anti-HEV IgM and negative for anti-HEV IgG, and two samples were seronegative, presumably because of a window period. Out of five donors reactive to HEV RNA at the initial testing, four were available for follow-up sampling and testing. One donor was withdrawn from further donations for reasons unrelated to HEV and was not available for follow-up testing. At follow-up, on average 6 months after the initial detection, HEV RNA was undetectable in all donors. At this time point, all blood donors had become positive for anti-HEV IgG, while only one donor, who had the highest level of viremia at the initial testing, remained positive for anti-HEV IgM (Table 3).

The ORF2 fragment was successfully amplified from only one plasma sample obtained from a donor from Belgorod. This sample had a high viral load, based on a Ct value of 20.9 (donor BR_2). The amplification failed in four other samples from viremic donors, presumably due to the low viral load. The phylogenetic analysis demonstrated that the identified HEV sequence (GenBank accession number BakIt2795048) belonged to HEV-3 sub-genotype 3a (Figure 1). Interestingly, the vast majority of HEV-3 sequences identified so far in Russia in general (shown in green in Figure 1) and in the Belgorod region (shown in red in Figure 1) belong to another HEV-3 clade that includes sub-genotypes e, f, and g. However, one HEV-3 sub-genotype 3a sequence of human origin was isolated previously in the Belgorod region (Figure 1), confirming the presence of this particular sub-genotype in the region.

## 4. Discussion

In this study, we investigated the proportion of blood donors with viremic HEV infection to assess whether HEV RNA testing should be implemented for blood services in Russia. Previous studies have shown that the HEV epidemiology in Russia is similar to that observed in many industrialized countries which are of intermediate HEV endemicity, with rather high seroprevalence due to asymptomatic infections in the general population and increased seroprevalence rates in seniors, and zoonotic HEV-3 is the causative agent of autochthonous cases [14]. Thus, although TT-HEV cases have not been reported in Russia so far, they are expected to exist but are unrecognized and/or undiagnosed. This is supported by the rather high detection rates of anti-HEV IgM antibodies in asymptomatic voluntary blood donors reported previously in the Moscow and Belgorod regions, at 2.8% and 4.5%, respectively [16], which equal the anti-HEV IgM rates in donors in countries that have already implemented HEV screening for their blood services [18]. 

Two particular regions, Moscow and Belgorod, were chosen for this study, because the average anti-HEV IgG prevalence rate in the general population in the former is the same as the country’s average (4%), but it is four times higher in the latter, reaching 16.4% [14], which is indicative of a higher intensity of virus circulation. Indeed, the ratio of HEV RNA positivity among blood donors appeared to be 1.5 times higher in Belgorod than in Moscow (1:2936 and 1:4844, respectively), although this difference was not statistically significant. Thus, we were able to combine the data from the two study regions to calculate the average ratio of HEV RNA positivity among donors, which was 1:4081. Considering that, with few exceptions, the prevalence of anti-HEV IgG in the majority of Russian regions, in both the European and Asian parts of the country, is similar to that of Moscow [14], we can expect that the average HEV viremia rate in blood donors observed in this study could be extrapolated to the whole country. However, studies in different parts of the country are needed to confirm this assumption. 

The HEV RNA prevalence of 0.02% observed in the donors in our study is similar to the rates reported in studies of donors in many countries with low/intermediate endemicity, where it is estimated to be 0.02–0.04% [19,20,21,22,23]. However, in several European countries, including Germany, France, and Serbia, HEV RNA positivity in donors was estimated to be significantly higher, at 0.18–0.31% [24]. Interestingly, the real-world data on HEV RNA prevalence in blood donors following the implementation of routine HEV screening for blood services might differ significantly from the data obtained in limited studies. For example, in England, HEV RNA in donors decreased from 1:1365 to 1:2848 donations in earlier studies to 1:4781 donations in routine universal screening using the same pool testing strategy [22]. Surprisingly, the HEV RNA prevalence rates among donors in highly endemic countries were reported to be similar to those observed in Europe. For example, Mishra et al., using MP-10 screening in India and an assay with a limit of detection of 4.7 IU/mL, identified 7 viremic cases out 13,050 donors, which amounts to 0.053%, or a rate of 1:1864 [25]. Similarly, the HEV RNA positivity rate was reported to be 0.02%, or 1:5000, among blood donors in Hong Kong in a study employing ID testing [26].

However, in some countries, where the virus is either endemic or non-endemic, HEV RNA was not detected in plasma samples from blood donors despite significant levels of seropositivity in the studied cohorts [27,28]. Indeed, differences in the frequency of detection of HEV viremia in donors can be explained by objective differences in the prevalence of HEV infection in the general population in a given territory. Thus, in a recent meta-analysis by Wolski et al., the risk of donors in North America being HEV viremic was estimated to be ten-fold lower compared to Europe, in line with the respective differences in seroprevalence in these regions of the world [24]. Notably, no HEV RNA-positive donations were identified in a study from Kazakhstan, when 16,147 donors, constituting 6.8% of the country’s donor population, were screened using the same test and the same pooling strategy as were used in our study [29]. However, the risk of HEV-3 and HEV-4 transmission in humans is largely related to diet, particularly the consumption of pork meat or by-products uncooked or partially cooked. Kazakhstan, being mostly a Muslim country, consumes and produces little pork, which might explain the very low prevalence of zoonotic HEV. However, differences in reported HEV viremia rates in donors from countries with a similar epidemiology appear to be largely related to sample size, the sensitivity of the test used, and the testing strategy, i.e., testing of individual donor samples or pools of different sizes. For the cobas^®^ HEV test that was employed in our study, which has a 95% limit of detection of 18.6 IU/mL, the expected proportion of HEV viremic donations that could possibly be missed due to testing in pools was recently calculated to be 15.2% for MP-6, 21.8% for MP-12, and 32.2% for MP-24 [30]. Thus, the MP-6 testing strategy employed in our study could potentially have led to an underestimation of HEV RNA prevalence in the donor cohort. The viral load in HEV reactive donations identified in our study is unknown, as the test was not a quantitative one, which can be considered a possible limitation of this study. The only available indirect reference in this case was PCR Ct values, which did not differ by more than four cycles between MP-6 and ID testing, indicating that only donations with very low viral load could remain undetected when tested in MP-6. Although the majority of TT-HEV cases are caused by blood components that contain 4 log IU/mL or more HEV RNA [12,31,32], donated blood with a lower virus content can pose a risk for immunocompromised patients, suggesting that there is a need to carry out ID testing of donations intended for this group of patients. 

Only one out of five HEV reactive donations identified in our study was confirmed by sequencing. The analysis of the only available HEV sequence isolated from the donor sample with the lowest Ct value confirmed the autochthonous nature of the HEV-3 infection in this donor. However, all donors available for follow-up testing seroconverted, which could be considered confirmation of previous infection. The absence of detectable anti-HEV antibodies in two out five viremic donors at the time of initial testing suggests the ineffectiveness of serologic donor screening. Therefore, the obtained results allow us to consider that introducing MP-6 HEV RNA screening of blood donors is an appropriate measure to ensure blood safety and minimize the risk of TT-HEV.

## 5. Conclusions

Our data demonstrate that the HEV RNA prevalence in blood donors in Russia corresponds with data from other European countries, including those that have implemented universal donor HEV screening. These data support the implementation of HEV RNA donor screening employing the MP-6 testing strategy to reduce the risk of transfusion-transmitted HEV infection in Russia.

## Figures and Tables

**Figure 1 viruses-16-00526-f001:**
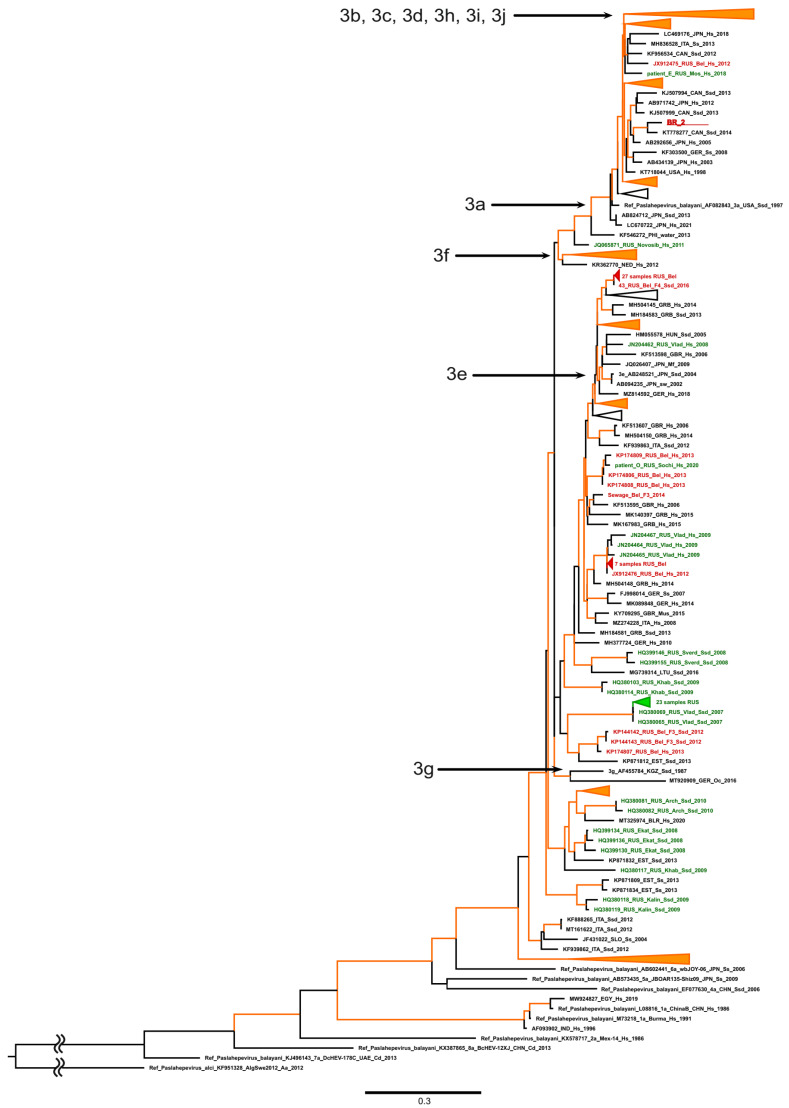
Maximum-likelihood phylogenetic tree based on 300 nt ORF-2 HEV sequences. Tree root was shortened to gain visibility. HEV-3 sub-genotypes (3a to 3j) are indicated with arrows. GenBank accession number, country (region), host organism, and year of isolation are indicated for reference sequences. Host designations are as follows: H.s., human (homo sapiens); Ssd, domestic pig (Sus scrofa domesticus); Ss, wild boar (Sus scrofa). Sequence from this study is shown in bold red and underlined. Other HEV sequences from Belgorod region are shown in red; sequences from other regions of Russia are shown in green. Tree branches with group reliability >90% are indicated in orange.

**Table 1 viruses-16-00526-t001:** Rates of HEV RNA in repetitive voluntary non-renumerated blood donors.

Study Region	Number of Tested Donors	Number of HEV RNA-Positive Donors	HEV RNA Detection Rate, % (95% CI)	HEV RNA Positivity Ratio	*p* *
Moscow	14,533	3	0.021% (0.01–0.06%)	1:4844	0.9518
Belgorod	5872	2	0.034% (0.01–0.13%)	1:2936	
Both regions combined	20,405	5	0.024% (0.01–0.05%)	1:4081	

* Chi-square with Yates correction when two regions were compared.

**Table 2 viruses-16-00526-t002:** Demographic characteristics of reactive donors and details of HEV RNA detection.

Donor #	Study Region	Donor Gender	Donor Age, Years	Ct in Pool of 6	Ct in Individual Samples
MSK_1	Moscow	female	39	40.4	37.5
MSK_2	Moscow	female	54	39.1	41.4
MSK_3	Moscow	male	42	40.5	37.1
BR_1	Belgorod	male	41	37.6	33.5
BR_2	Belgorod	male	38	25.0	20.9

**Table 3 viruses-16-00526-t003:** Donor HEV RNA and anti-HEV antibody status at initial and follow-up testing.

Donor ID	Initial Testing			Follow-Up Testing (6 Months Later, on Average)		
	HEV RNA	Anti-HEV IgM	Anti-HEV IgG	HEV RNA	Anti-HEV IgM	Anti-HEV IgG
MSK_1	positive	positive	negative	lost to follow-up	no data	no data
MSK_2	positive	positive	positive	negative	negative	positive
MSK_3	positive	negative	negative	negative	negative	positive
BR_1	positive	positive	positive	negative	negative	positive
BR_2	positive	negative	negative	negative	positive	positive

## Data Availability

The data presented in this study are available in this article.

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
