# Peer review of "Hepatitis E Virus Infection in Voluntary Blood Donors in the Russian Federation"

_viruses, 2024, doi:10.3390/v16040526_

Round 1
Reviewer 1 Report
Comments and Suggestions for Authors
The manuscript from Potemkin IA et al reports a medium size blood donor screening study for HEV RNA. It shows a relatively low prevalence of window-period/early infections.
In materials and methods, it would be useful to indicate the limit of detection (LOD) of the Roche HEV assay in both individual and pools of 6 samples instead of mentioning it in the discussion. It might also be useful to indicate that Belgorod is located about 600 km south of Moscow, close to the Ukraine border.
In the discussion, please indicate that the Mishra study was conducted in India.
The authors should mention that to a large extent, the epidemiology of HEV genotypes 3 and 4 are related to diet, particularly eating pork meat uncooked or partially cooked such as in sausage and charcuterie. Kazakhstan being mostly a Muslim country do not consume pork that might explain very low prevalence of HEV genotype 3.
Comments on the Quality of English LanguageQuite good with minor improvements helpful.
Author Response
We are very grateful to Reviewer for comments and thorough analysis of our paper.
Comment 1
In materials and methods, it would be useful to indicate the limit of detection (LOD) of the Roche HEV assay in both individual and pools of 6 samples instead of mentioning it in the discussion. It might also be useful to indicate that Belgorod is located about 600 km south of Moscow, close to the Ukraine border.
Response
According to kit insert, the limit of detection (LoD) of the test is 18.6 HEV RNA International units (IU) per mL (95% CI: 15.9-22.6 IU/mL) when samples tested individually. To our best knowledge, LoD of the assay for mini-pools of 6 was never assessed in any published study, but it can be deduced approximately from LoD data for individual samples (18.6 x 6, 2.05 log10/mL). We added this information to methods, lines 105-107 in revised manuscript.
We also added the information on Belgorod location in Introduction, lines 72-73 in revised manuscript.
Comment 2
In the discussion, please indicate that the Mishra study was conducted in India.
Response
Yes, this study was conducted in India. We added this information (line 227 in revised manuscript).
Comment 3
The authors should mention that to a large extent, the epidemiology of HEV genotypes 3 and 4 are related to diet, particularly eating pork meat uncooked or partially cooked such as in sausage and charcuterie. Kazakhstan being mostly a Muslim country do not consume pork that might explain very low prevalence of HEV genotype 3.
Response
We do agree that HEV genotypes 3 and 4 transmission in humans is largely related to diet. Thus, the risk of these HEV genotype acquisition in Muslim countries is limited, which may be a possible explanation of very low HEV prevalence in Kazakhstan. We added this point to Discussion, lines 241-244 in revised manuscript.
Reviewer 2 Report
Comments and Suggestions for Authors
The manuscript titled "Hepatitis E virus infection in voluntary blood donors in the Russian Federation" by Ilya A. Potemkin et al. reports the results of a screening for HEV-RNA in blood donors from two regions of Russian federation.
The results confirm a low prevalence of HEV-RNA.
The results seem interesting and could widen the knowledge gap related to the spread of HEV infection and the risk of transfusion-transmitted HEV infection. However, it is my opinion that the manuscript requires some modifications before being accepted in the journal Viruses.
For these reasons, I suggest the authors consider the following comments.
L. 45-46. Provide some information on HEV-5, -6, -7, and -8 genotypes by indicating the possibility that HEV-7 is also zoonotic (see: doi: 10.1053/j.gastro.2015.10.048)
L. 47. Delete "spillover". HEV-3 and HEV-4 infection is endemic in pigs and humans in many industrialized countries. Also suggest specifying that in humans (and animals) most infections are asymptomatic.
L. 80-82. It is not clear to me whether, (a) the 20.405 samples tested are all from different blood donors or, (b) in some cases these are different blood donations from the same blood donor. If hypothesis (b) is true, I suggest specifying from how many blood donors the blood donations came from.
I also suggest stating: the number of donations/donors of male and female gender; median and age range of males; median and age range of females; start and end date of the study.
L. 109. Replace "Subsection" with something like "HEV sequencing and Phylogenetic analyses"
L. 122. In this part of the text the authors write "prevalence" while in others they write "detection rate." If the data refer to blood donors I think it is more correct to use the term "prevalence" (see my previous comment).
L. 139. Delete the sentence "The median age of HEV RNA positive donors was 41 years." The age of positive individuals is shown in Table 2.
Table 3. Replace "no data" with "Loss to follow-up."
L. 181-184. I think this sentence is contradictory: the authors first write that HEV is considered non-endemic and immediately after that the seroprevalence is quite high. I recommend rephrasing the sentence.
Author Response
Reviewer 2
We are grateful to the Reviewer for the comments and thorough analysis of the manuscript.
Comment 1
L. 45-46. Provide some information on HEV-5, -6, -7, and -8 genotypes by indicating the possibility that HEV-7 is also zoonotic (see: doi: 10.1053/j.gastro.2015.10.048)
Response
We added information on HEV genotypes 5 to 8 (lines 49-51 in revised manuscript).
Comment 2
L. 47. Delete "spillover". HEV-3 and HEV-4 infection is endemic in pigs and humans in many industrialized countries. Also suggest specifying that in humans (and animals) most infections are asymptomatic.
Response
We deleted "spillover" and specified that HEV-3 and HEV-4 infections in animals and immunocompetent humans are often asymptomatic and self-limited (lines 47-48 in revised manuscript).
Comment 3
L. 80-82. It is not clear to me whether, (a) the 20.405 samples tested are all from different blood donors or, (b) in some cases these are different blood donations from the same blood donor. If hypothesis (b) is true, I suggest specifying from how many blood donors the blood donations came from.
Response
All samples were obtained from different donors, i.e. single sample was obtained from each donor. We specified this in Methods section (lines 88-90 in revised manuscript).
Comment 4
I also suggest stating: the number of donations/donors of male and female gender; median and age range of males; median and age range of females; start and end date of the study.
Response
Unfortunately, demographic data (gender and age) of donors who participated the study are not available. All donors were adults (18 years and older). The sample collection was performed from October 2022 to September 2023. We added this information to Methods section (line 97 in revised manuscript).
Comment 5
L. 109. Replace "Subsection" with something like "HEV sequencing and Phylogenetic analyses"
Response
Thank you for noticing the missed subsection title. We used the suggested title for this subsection.
Comment 6
L. 122. In this part of the text the authors write "prevalence" while in others they write "detection rate." If the data refer to blood donors I think it is more correct to use the term "prevalence" (see my previous comment).
Response
Indeed, the data refer to donors, so we use term "prevalence" instead of "detection rate" throughout the text.
Comment 7
L. 139. Delete the sentence "The median age of HEV RNA positive donors was 41 years." The age of positive individuals is shown in Table 2.
Response
We deleted this sentence.
Comment 8
Table 3. Replace "no data" with "Loss to follow-up."
Response
We replaced "no data" with "loss to follow-up".
Comment 9
L. 181-184. I think this sentence is contradictory: the authors first write that HEV is considered non-endemic and immediately after that the seroprevalence is quite high. I recommend rephrasing the sentence.
We rephrased the sentence to state the intermediate HEV endemicity (line 193 in revised manuscript).